# Cues to improve antibiotic-allergy registration: A mixed-method study

**Martijn Sijbom**[1]*, **Karolina K. Braun**[1], **Frederike L. Büchner**[1], **Leti van Bodegom-Vos**[2], **Bart J. C. Hendriks**[3], **Mark G. J. de Boer**[4], **Mattijs E. Numans**[1], **Merel M. C. Lambregts**[4]

**1** Department of Public Health and Primary Care Campus-Den Haag, Leiden University Medical Center, Leiden, The Netherlands, **2** Department of Biomedical Data Sciences, Leiden University Medical Center, Leiden, The Netherlands, **3** Department of Clinical Pharmacy and Toxicology, Leiden University Medical Center, Leiden, The Netherlands, **4** Department of Infectious Diseases, Leiden University Medical Center, Leiden, The Netherlands

\* m.sijbom@lumc.nl

## Abstract

### Background

Approximately 2% of patients in primary care practice and up to 25% of hospital patients are registered as being allergic to an antibiotic. However, up to 90% of these registrations are incorrect, leading to unnecessary prescription of 2nd choice antibiotics with the attendant loss of efficacy, increased toxicity and antibiotic resistance. To improve registration, a better understanding is needed of how incorrect labels are attributed.

### Objective

To investigate the quality of antibiotic allergy registration in primary care and identify determinants to improve registration of antibiotic allergies.

### Design

Registration of antibiotic allergies in primary care practices were analysed for 1) completeness and 2) correctness. To identify determinants for improvement, semi-structured interviews with healthcare providers from four healthcare domains were conducted.

### Participants

A total of 300 antibiotic allergy registrations were analysed for completeness and correctness. Thirty-four healthcare providers were interviewed.

### Main measures

A registration was defined as complete when it included a description of all symptoms, time to onset of symptoms and the duration of symptoms. It was defined as correct when the conclusion was concordant with the Salden criteria. Determinants of correct antibiotic allergy registrations were divided into facilitators or obstructers.

**Data Availability Statement:** Our study had a mixed method design and was based on two datasets. A set with coded routine Medical Record data, pseudonymized and extracted from primary

care practices towards the ELAN datawarehouse, was used for the qualitative analysis of antibiotic allergy registrations. This dataset cannot be shared in an open public repository. Medical data in the ELAN datawarehouse are pseudonymized, so theoretically patients still can be identified and confidentiality could be violated. Patients did consent to reuse their medical data for the purpose of dedicated and contextually restricted research and quality management, but not in an open and publicly available domain. So, data are available upon reasonable request at the ELAN datawarehouse through < elanresearch.nl >. The dataset on determinants for incorrect antibiotic allergies registrations cannot be shared in a public repository for other reasons. These data are based on verbatims produced from audio recordings of interviews with research participants. We do not have consent to disseminate the full transcripts and we have to respect the anonymity of the research participants. Although anonymized, participants could be identified through their answers and confidentiality could be violated. Data is available upon reasonable request by emailing our data manager, Laura Haakmeester; l.a. haakmeester@lumc.nl.

**Funding:** The author(s) received no specific funding for this work.

**Competing interests:** The authors have declared that no competing interests exist.

## Key results

Rates of completeness and correctness of registrations were 0% and 29.3%, respectively. The main perceived barriers for correct antibiotic allergy registration were insufficient knowledge, lack of priority, limitations of registration features in electronic medical records (EMR), fear of medical liability and patients interpreting side-effects as allergies.

## Conclusions

The quality of antibiotic allergy registrations can be improved. Potential interventions include raising awareness of the consequences of incomplete and the importance of correct registrations, by continued education, and above all simplifying registration in an EMR by adequate ICT support.

## Introduction

Allergies to antibiotics are among the most commonly reported adverse reactions to medication. Adequate registration of these allergies is essential to prevent rare but potentially life-threatening reactions upon re-exposure. In Dutch primary care, 0.6% to 2.1% of patients have an antibiotic allergy registration in their electronic medical record (EMR) [1, 2]. Worldwide higher rates of antibiotic allergy registrations have been reported, ranging up to 25% [3]. However, between 80 to 90% of antibiotic allergy registrations in primary care are incorrect [1, 4, 5].

Antibiotic allergy registrations are associated with more frequent visits to the doctor, higher healthcare costs and more frequent prescription of second-choice antibiotics [2, 6–8]. Importantly, the efficacy and/or toxicity profiles of second-choice antibiotics are generally less favourable compared to the narrow spectrum antibiotics that most often constitute first choice of treatment. The use of broad-spectrum antibiotics also increases risk of *Clostridiodes difficile*-associated diarrhoea and promotes the emergence of antimicrobial resistance [9].

In the Netherlands antibiotic allergies are registered in all healthcare domains, including primary care, hospitals, pharmacies and long-term elderly care facilities. Primary care physicians play a pivotal role in the registration of antibiotic allergies, since in the Netherlands they function as gatekeeper for entry to most other healthcare fields. Ninety percent of antibiotic prescriptions, and the majority of antibiotic allergy registrations, originate in primary care [10]. EMRs kept in primary care contain all essential medical data and function as a central medical record for most other healthcare domains. Antibiotic allergies registered in other healthcare domains are subsequently recorded in the patient's primary care EMR and vice versa, thus facilitating further dissemination of antibiotic allergy registrations from one healthcare setting to the other. The registration of antibiotic allergies transcends primary care practice. Therefore, any effort to tackle this issue should be collaborative and involve all relevant healthcare domains.

Although the quality of current antibiotic allergy registration is known to be insufficient [1, 7, 8, 11], detailed insight into the specific aspects of registration that could be improved is lacking. In addition, a better understanding of the determinants of incorrect antibiotic allergy registration and -in particular- the similarities and differences between healthcare domains is needed. This information will be essential to the effective design and implementation of interventions aimed at improving antibiotic allergy registration.

The primary goals of this study were to analyze the quality of antibiotic allergy registrations in primary care and to identify determinants related to the quality of registration in all involved healthcare domains.

## Methods

### Study design

The study consisted of a point prevalence analysis of the quality of antibiotic allergy registrations in primary care, together with a qualitative study based on semi-structured interviews to assess the determinants of incorrect registration. Before the start of this study, the study was approved by the institutional Ethics Review Board of the Leiden University Medical Center (file number G19.007).

### Analysis of the quality of antibiotic allergy registrations in primary care

**Data collection.** Patient data were obtained through the Extramural LUMC Academic Network (ELAN), which includes 31 primary care practices in the Leiden-The Hague area and holds primary care data of approximately 200,000 patients. Primary care physicians involved in this network provide access to their anonymized EMRs medical data, that are accessible through the ELAN datawarehouse.

Antibiotic allergy registrations were identified based on the following registrations in the EMR: International Classification of Primary Care version 1 (ICPC) code A12 (allergy/allergic reaction) *or* A85 (adverse event medical agent) *or* a registration for a contraindication (CIA) label antibiotic allergy for Anatomical Therapeutic Chemical (ATC) code J01 (antibacterials for systemic use). The EMR in primary care supports registration of all relevant details within the allergy label, including symptoms and time course of the reaction. All registrations dated up until the year 2018 were used.

EMRs from primary care and pharmacies are linked and exchange information on antibiotic allergies automatically. The primary care antibiotic allergy label is not electronically linked to the EMR in hospitals nor long term care facilities. Information on allergy labels between primary care and hospitals/long term care facilities is exchanged through referral letters.

**Quality analysis of the allergy registration.** Quality analysis consisted of an assessment on completeness and correctness of the antibiotic allergy registration in the primary care EMR based on a previously published checklist by Salden et al. (S1 Table) [1]. The checklist was modified for one item: the maximal time between start of symptoms and first intake of antibiotic was extended to up to 6 hours for immediate type allergies (See Box 1, Immediate type versus delayed type antibiotic allergy). Assessment was conducted with information available in the registration. A complete registration was defined as a registration that contained a

---

Box 1. Immediate type versus delayed type antibiotic allergy

Immediate type allergies are IgE mediated reactions. The symptoms are the result of immediate release of histamine and other cytokines upon exposure to an allergen. The most frequently reported symptoms are urticaria, angio-oedema, exanthema, dyspnoea and hypotension, and occur within a few hours. This is opposed to delayed type reactions, which generally develop a few days after exposure, as they are cell-mediated. A mild exanthema is the most frequent delayed type reaction.

description of symptoms <u>and</u> time to onset of symptoms <u>and</u> duration of symptoms. Antibiotic allergy registrations were then classified as an 'immediate type reaction' (possible/probable), 'delayed type reaction' (possible/probable), 'non-allergic side effect' or 'insufficient data available for diagnosis'. A correct antibiotic allergy registration was defined as a registration in which the conclusion was concordant with the diagnosis according to the modified checklist.

To represent daily practice, analysis of antibiotic allergy registrations was limited to the 5 antibiotic groups most frequently prescribed in primary care in the Netherlands: penicillins, tetracyclines, nitrofuran derivatives (i.e. nitrofurantoin), macrolides and fluoroquinolones [10]. A sample of 300 antibiotic allergy registrations was obtained for quality analysis. The size of the random sample was calculated using a random sample formula [12]. We used a confidence level of 90% and a margin of error of 5%, including the entire ELAN data warehouse population for each type of registration. These 300 patients were selected through randomisation by SPSS (version 25, SPSS Inc., Chicago, IL). If a patient had multiple antibiotic allergy registrations, 1 registration was randomly selected and used for further analysis.

## Statistical analysis

Analyses were conducted using SPSS, version 25. The prevalence of patients with an antibiotic allergy registration was calculated for all registrations and for the 5 most frequently prescribed antibiotics groups. Unpaired t-tests were applied to compare continuous variables with normal distributions and reported as a 95% confidence interval (95% CI). Age was reported as a median and with an interquartile range (IQR).

## Determinants of correct antibiotic allergy registrations

**Semi-structured interviews.**   To identify determinants of correct antibiotic allergy registration, five interviewers (KB, ML, YA, BH and MS) conducted semi-structured interviews with primary care, hospital care, elderly care and pharmacy healthcare workers in the Leiden and The Hague regions of the Netherlands. This region encompasses a large metropolitan area. This part of the study was conducted and reported according to the Consolidated Criteria for Reporting Qualitative Research (COREQ) checklist (S2 Table) [13].

Participants were selected using a purposive sampling method to represent the healthcare workers in the region who encounter antibiotic allergy registrations, taking into account differences in experience and sex and asked to participate via e-mail or face-to-face [14].

The semi-structured interview (S3 Table) contained questions based on themes from a checklist by Flottorp et al. [15]. This checklist describes themes that obstruct or facilitate improvements in healthcare: guideline factors, individual healthcare professional factors, patients factors, professional interaction, incentives and resources, capacity for organisational change, and social, political and legal factors.

A pilot interview was performed and followed by semi-structured interviews that were conducted until saturation of answers occurred, with a minimum of 10 interviews [14]. Saturation was defined as no new information in 3 consecutive interviews. At saturation, answers were considered to give a complete overview of all possible answers.

All interviews were digitally recorded after obtaining permission from interviewees and transcribed verbatim. Transcripts were uploaded in Atlas.Ti, version 8, and coded. A three-step plan was used for content analysis. The first step consisted of labelling individual quotes. In step 2, labels were coded by theme. In the third and final step, labelled quotes were identified and coded per determinant, and then categorised as either facilitator and barrier. Two researchers (K.B, M.S.) independently performed the coding. Any discrepancies in coding were resolved by discussion. If consensus could not be reached, a third reviewer was asked to

resolve any outstanding issues (F.B.). The identified determinants were structured into a framework according to the themes in the checklist of Flottorp.

## Results

### Analysis of the quality of antibiotic allergy registrations in primary care

The ELAN data warehouse contained routine registry data on 196,038 enlisted patients (0–102 years) at the time of analysis. The prevalence of registered patients with an antibiotic allergy registration was 3.2% (6368/196,038), encompassing 11,841 antibiotic allergy registrations in total (Table 1). Of the 6368 patients with an antibiotic allergy registration, 2034 had multiple registrations, ranging from 2 to 22 per patient. Penicillin allergy was the most frequently registered antibiotic allergy, 45.0% (95% CI from 44.1% up to 45.9%).

Assessment of 300 antibiotic allergy registrations using the modified Salden checklist showed that none of these registrations were complete (Table 2). Information on the time course of symptoms were missing in 80% of cases. According to the Salden criteria, diagnosis of an antibiotic allergy was correct in 29.3% (n = 88/300) of registrations (Table 3). In 14.3% (n = 43/300) of cases, a non-allergic reaction was incorrectly registered as an antibiotic allergy.

### Semi-structured interviews

In total, 31 primary care physicians (PCP), 4 medical specialists (MS), 11 Elderly Care physicians (ECP), 5 elderly care nurses (ECN) and 4 Pharmacists or pharmacy technicians (PH) were invited to participate. Data saturation was reached after interviews with 10 PCPs, 4 MSs, 11 ECPs, 5 ECNs and 4 PHs, of whom 56% was female and 53% had more than 10 years' experience. The MS consisted of a surgeon in training, a hospital physician and 2 gastroenterologists. Transcripts were analysed according to the 3-step plan described in the methods (Fig 1 and Table 4).

### Individual characteristics of care providers

All healthcare providers stated that side effects were sometimes registered as allergies, with the interviewees explaining that side effects were interpreted as allergies either due to lack of knowledge, medical uncertainty and/or fear of medical liability. In all domains, healthcare

**Table 1. Characteristics of patients with an antibiotic allergy registration.**

|  | Cohort of patients with an allergy registration | Random selection of 300 allergy registrations |
|---|---|---|
| Patients (n) | 6368 | 300 |
| Patients with multiple registrations (n) | 2034 | 0 |
| Sex % female (n) | 73.1% (4655) | 73.3% (220) |
| Age at diagnosis of first antibiotic allergy registration (min-max years) | 0–102 (median 51 years, IQR 31–68 years) | 0–98 years (median 50 years, IQR 32–67 years) |
| Antibiotic allergy registrations (n) | 11,841 (100%) | 300 (100%) |
| Penicillins % (n) | 45.0% (5323) | 61.3% (184) |
| Tetracyclines % (n) | 7.7% (912) | 10.0% (30) |
| Nitrofuran derivatives % (n) | 10.3% (1224) | 16.7% (50) |
| Macrolides % (n) | 6.7% (793) | 8.0% (24) |
| Fluoroquinolones % (n) | 5.4% (641) | 4.0% (12) |
| Other % | 24.9% (2948) | 0 (0) |

95% CI, 95% confidence interval; IQR, Interquartile range.

**Table 2. Analysis of a random selection of antibiotic allergy registrations for completeness and correctness.**

| Noted in registration | Total (n = 300) |
|---|---|
| Registration of substance* | 93.7% (281) |
| Time to start of symptoms† | 20% (60) |
| Duration of symptoms‡ | 7.3% (22) |
| Description of symptoms§ | 46.3% (139) |
| Hospital admission‖ | 0% (0) |
| Allergy test¶ | 0% (0) |
| Prescribed again# | 20.3% (61) |
| Type of allergy** | 0% (0) |

*Antibiotic was specified in registration.

†Time between first intake of antibiotic and start of symptoms.

‡Duration of symptoms after first intake of antibiotic.

§Description of symptoms present in registration.

‖Registration of whether hospital admission was needed to treat antibiotic allergy reaction.

¶Registration of whether an allergy test was performed.

#Antibiotic for which an allergy was registered was prescribed again after registration.

**Type of allergic reaction was specified in registration: immediate versus delayed type.

providers admitted a lack of knowledge regarding distinguishing side effects from various types of antibiotic allergies. Interviewees who were aware of the issue of incorrect antibiotic allergy registrations, were more likely to verify existing registrations. They also indicated that these processes require education concerning antibiotic allergies and expressed a wish for more educational opportunities.

## Patient factors

Patient factors, such as cognitive impairment or aphasia, hinder verification and classification of previously registered allergies. This problem was mentioned in particular by ECPs. According to interviewees, the patient's preferences and personal interpretation of symptoms lead to incorrect registrations. Patients sometimes prefer not to be prescribed a specific antibiotic based on previous experiences, i.e. side-effects. This can lead to incorrect antibiotic allergy registration, but prevents patient exposure to the antibiotic.

## Professional interactions

Interviewed PCPs reported hardly any problems regarding communication of antibiotic allergies with other healthcare providers both ways, stating that most communication was digital

**Table 3. Type of allergic reaction according to modified checklist of Salden\*.**

| Type of reaction | Total (n = 300) |
|---|---|
| Immediate type reaction probable | 0% (0) |
| Immediate type reaction possible | 2.0% (6) |
| Delayed type reaction probable | 0% (0) |
| Delayed type reaction possible | 18.3% (55) |
| No distinction possible between immediate or delayed reaction | 9% (27) |
| No allergic reaction | 14.3% (43) |
| Type of reaction could not be determined | 56.3% (169) |

*Information in registrations was compared to modified checklist of Salden, see S1 Table for details.

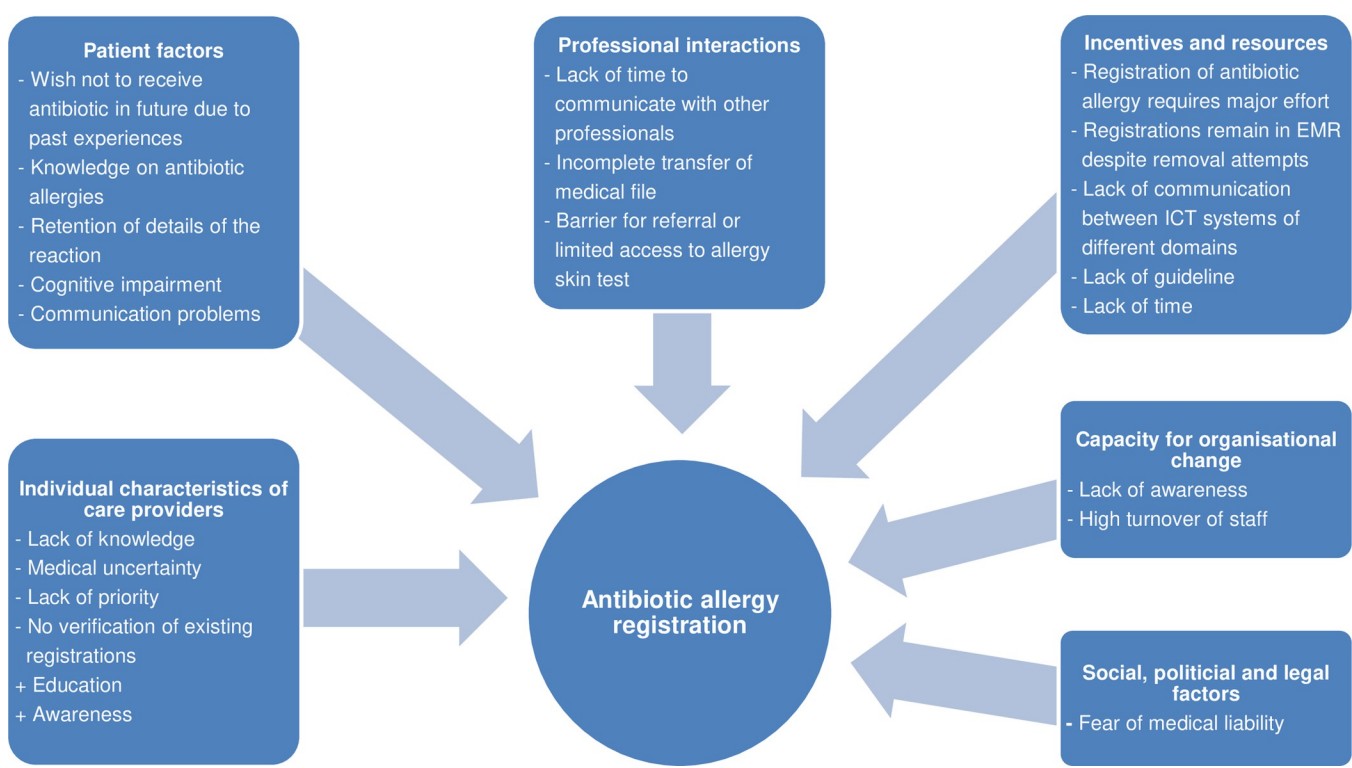

**Fig 1. Determinants of antibiotic allergy registration.** + = facilitator;— = barrier; EMR: Electronic Medical Record; ICT: Information Communication Technology.

through their EMRs and was sufficient in their opinion. Interviewed PCPs also mentioned that more elaborate communication was mainly confined to pharmacists but was hindered by lack of time. Other healthcare providers occasionally experienced difficulties in communication, stating that EMR registrations were sometimes incomplete, referral letters were missing essential details. Reaching other healthcare providers to obtain missing information was time-consuming. Together, these issues made it difficult to verify an antibiotic allergy registration. According to PCPs, another barrier for correct registration of antibiotic allergies was limited availability or access to diagnostic tests, in addition to (presumed) long waiting lists for referral to an allergist.

## Incentives and resources

Lack of time hindered complete and correct registration of new antibiotic allergies. Furthermore, lack of time often led to healthcare providers failing to verify whether an existing antibiotic allergy registration was correct.

Many different EMR systems are in use in the Netherlands. According to interviewees, all EMR systems presented greater or lesser difficulties when registering a reaction, and EMR systems did not support a clear distinction between a side effect/ intolerance and allergy. Both registration of a new allergy and retrieval of information on previously reported allergies is time consuming. Interviewees mentioned that miscommunication between different EMRs resulted in missing information and hindered removal of incorrect antibiotic allergy registrations.

None of the interviewed healthcare providers used a protocol or specific procedure for registering antibiotic allergies, although some expressed a wish for a guideline. According to the

**Table 4. Examples of quotes per determinant.**

| DETERMINANTS | QUOTE |
|---|---|
| **INDIVIDUAL CHARACTERISTICS OF HEALTHCARE PROVIDERS** | |
| Lack of knowledge | PH2: *"Yes, interpreting a complaint as an allergy is sometimes quite difficult: when is it really an allergy? And a cross-sensitivity, I don't think our technicians can handle that."* |
| Medical uncertainty | PCP6: *"When do you call something an allergy and when a side effect? Urticaria is both a side effect and an allergy. Both are plausible, so how do I choose?"* |
| Lack of priority | PCP4: *"In my experience it [antibiotic allergy] doesn't happen often and isn't necessarily relevant."* |
| No verification of existing registrations | MS1: *"Because I don't think everyone checks [the registration] with the patient. And also because it is of course easier to simply copy the information from your predecessor [previous healthcare provider]."* |
| Education | PH1: *"It would be nice if, perhaps—I think if we were better educated [in antibiotic allergies], we would be better at registering it."* |
| Verification | PH2: *"We say to patients: 'Our EMR says that you're hypersensitive or allergic to amoxicillin or penicillin. [. . .] Is that correct?' and the patient answers yes or no. If the answer is yes, I always ask about the symptoms because they actually determine whether or not someone is really allergic".* |
| Awareness | MS1: *"Actually, forgive me for saying this, but when I see a registration I don't entirely trust it because I often find that it is not quite right. That's why I always verify it for myself."* |
| **PATIENT FACTORS** | |
| General knowledge on antibiotic allergy | PCP3: *"Patients see side effects as allergies."* |
| Retention of details of the reaction | PCP1: *"Patients, in general, have a bad memory for [names of] medication."* |
| Wish not to receive antibiotic due to past experience | PCP1: *"I think the patient's opinion is important. So when they say that they don't want the first choice antibiotic because of side effects, I look at what else I can offer."* |
| Cognitive impairments | ECP7: *"The psycho-geriatric residents, who are in a poor mental state, can't be asked about when and what kind of allergy they may have had. That's not possible."* |
| Communication problems | ECP4: *". . . [The psycho-geriatric residents] have problems with aphasia, of course. Or poor vision or hearing. . . ."* |
| **PROFESSIONAL INTERACTIONS AND REPORTING** | |
| Lack of time to communicate | PCP1: *"In itself, I think that communication is fine, but in practical terms it is difficult because we have many patients and only limited time."* |
| Incomplete transfer of medical file | ECP16: *"And if I thoroughly read the doctor's notes, it often just says somewhere " 2014 allergy to amoxicillin" but doesn't provide any more detail."* |
| Barrier for referral or limited access to allergy skin test | PCP9: *"But these allergists have long waiting lists, though I don't know how long. So if you need acute assistance, that doesn't help either."* |
| **INCENTIVES AND RESOURCES** | |
| Registrations remain in the EMR | PCP1: *"Once you put it in the system, it can't be easily removed. And because it requires a lot of effort to determine if it is justified, the next doctor will just see it and act on it."* |
| Lack of time | MS2: *"I also think that we are not really aware of the problem [incorrect antibiotic allergies]. So much has to be done at the same time. It is just busy."* |
| Registration in EMR takes effort | PH2: *"You know, these are things you can't justify spending that amount of time on. Especially if you have to figure it all out."* |
| Communication between ICT systems | PCP7: *"The [allergy] is registered at the pharmacy, but it isn't in my system."* |
| Lack of guideline | ECP12: *"If everyone would register in the same way seems useful to me, so yes."* |

*(Continued)*

**Table 4.** (Continued)

| DETERMINANTS | QUOTE |
|---|---|
| **INDIVIDUAL CHARACTERISTICS OF HEALTHCARE PROVIDERS** | |
| **CAPACITY FOR ORGANISATIONAL CHANGE** | |
| Lack of awareness | PCP1: *"Theory is fun, but awareness is more important. You shouldn't oblige people to immerse themselves in the subject because that isn't feasible. PCPs are already driven mad by everything they have to do."* |
| High turnover of staff | ECP2: *"An ongoing obstacle is the fairly rapid turnover of staff, because if you think 'Okay, I'm going to do it correctly', new staff turn up and you have to start all over again. "* |
| **SOCIAL, POLITICAL AND LEGAL FACTORS** | |
| Medical liability | MS2: *"Yes, sometimes it's not entirely clear. What should I do next? Just to be sure, you give them something else."* |

ECP, Elderly Care Physician; PCP, Primary Care Physician; PH, Pharmacist; MS, Medical specialist; EMR, Electronic Medical Record; ICT, Information Communication Technology.

interviewees, a guideline should be accompanied by a decision support system in an EMR and together these were seen as an effective solution.

## Capacity for organizational change

Incorrect antibiotic allergy registrations were not deemed to be problematic by PCP's and hence they gave little priority to improving the verification of existing antibiotic allergies. They stated there is "*no need as there is always an alternative antibiotic available*". In contrast, ECPs more frequently perceived allergy registrations as a problem as they frequently encountered patients with multiple antibiotic allergy registrations, hindering the selection of an appropriate antibiotic. An ECP also commented that high staff turnover impeded the necessary changes in policy to ensure correct registration of antibiotic allergies.

## Social, political and legal factors

One interviewee also stated that, based on previous personal experience, fear of medical liability can lead to incorrect registration of antibiotic allergies or omission to remove a previous registration.

## Discussion

The main finding of our study is that in the majority of cases (56.3%) recorded information was insufficient to determine whether the reaction was of an allergic nature. Main causes of insufficient quality of registrations were lack of knowledge, lack of priority, limitations of registration features in EMRs and patients interpreting side–effects as allergies.

### Analysis of the quality of antibiotic allergy registrations in primary care

Our study provides detailed new insight into what is lacking in antibiotic allergy registrations. In our quality assessment, non-allergic reactions interpreted as antibiotic allergic reactions accounted for 14.3% of all registrations, a figure comparable to the 11.7% reported by Salden et al. [1]. This is however an underestimate of the actual number of reactions that are incorrectly labelled as an allergy: 56.3% of antibiotic allergy registrations lacked essential information such as a description of symptoms, their time of onset and/or duration. Such detailed

information is needed in order to determine the type and severity of the reaction and to be able to decide whether an antibiotic can be prescribed safely.

Although delayed type reactions cause discomfort, they are rarely life-threatening except in very rare cases such as Stevens-Johnson syndrome or toxic epidermal necrolysis (SJS/TEN) and drug rash with eosinophilia and systemic symptoms (DRESS). Risk of recurrence of a mild delayed type reaction is low and there is no additional risk of an immediate type reaction with the exception of severe cutaneous adverse reactions [16]. Therefore, a mild delayed type reaction would not be an absolute contra-indication for the antibiotic in question. To be able to decide on re-exposure, a complete antibiotic allergy registration is needed. When the details of the reaction can't be retrieved, for example if the patient does not remember and there is no documentation, this should be indicated in the EMR.

## Determinants of incorrect antibiotic allergy registration

Health care providers' lack of knowledge regarding the differentiation of allergic versus non-allergic reactions was perceived as a major determinant of incorrect registration. Similar findings were reported in one primary care study and two studies of hospital doctors [17–19]. Improved education of healthcare providers registering antibiotic allergies is a possible solution to overcome incorrect interpretations.

Interviewees from all domains perceived patient related factors as important determinants of incorrect antibiotic allergy registrations. Firstly, patients may not remember the details of the reaction, especially if the reactions occurred in remote childhood. Secondly, patients may interpret side effects as an allergy and express a wish not to receive a particular antibiotic in the future, often resulting in the incorrect registration of an antibiotic allergy. A study by De Clercq et al. reported similar findings in primary care [17]. Interviewees also stated that a clear explanation and effective communication with the patient can help to avoid an incorrect registration. Patient-orientated research in which patients are interviewed concerning their experiences of side effects and antibiotic allergic reactions is needed to gain more insight into this particular determinant. These findings might then be used to design and implement patient-directed interventions.

In this study, unawareness of the problem of incorrect antibiotic allergy registration and its consequences was an issue in all healthcare domains, especially in primary care. While most PCPs were unaware of the problem of incorrect registration of allergies, ECPs by contrast regularly encountered patients with multiple antibiotic allergy registrations, severely hindering the prescription of the correct antibiotic. Multiple antibiotic allergy registrations are most likely the result of lifelong collection of registrations. The lack of awareness is concordant with earlier reports in primary and hospital care and suggests that greater awareness is needed to change the behaviour of healthcare providers [6, 7, 20]. In a study by Schouten et al., improved awareness played a key role in removing barriers to optimal antibiotic therapy in a hospital setting [21]. Interventions to improve antibiotic allergy registrations should therefore focus not only on improving knowledge but also on increasing awareness.

Another important perceived determinant was the failure of EMR software to support the quick and accurate registration of symptoms and their time-course. EMR software developers need to simplify registration and allow a distinction between allergy or side effect [17].

Some interviewees suggested development of a guideline accompanied by a clinical decision making system in the EMR. A study by Blumenthal et al. showed that this type of system can indeed improve the registration of antibiotic allergies in a hospital setting [22]. Most incorrect antibiotic allergy registrations can be safely removed with a thorough history with or without a provocation test [23]. In most cases skin testing is not needed. Guidelines on the clinical

approach of a potential antibiotic allergy and removing of incorrect antibiotic allergies are highly needed.

To a greater or lesser extent, domains mostly shared the same determinants. This supports the development of interventions that transcend the individual healthcare domains. For example, educational programs may be developed targeting all domains, with the aim to improve knowledge, but also interdisciplinary communication and collaboration. Furthermore, ICT registration and decision tools could be developed to support both primary care and hospital care.

## Validity and limitations

A strength of our quality analysis was the use of routinely registered medical data from primary care. This data reflects daily practice regarding the registration of antibiotic allergies.

A strength of our interviews was the inclusion of healthcare workers from all domains that register antibiotic allergies, hence providing a complete overview. A comprehensive approach is important as antibiotic allergy registrations clearly transcend the individual domains. The relevance is illustrated by the determinants that were identified regarding the interactions between healthcare domains and individual healthcare professionals.

An advantage of semi-structured interviews is that it allows an interviewer the freedom to pursue more in-depth answers to specific questions, without compromising the comparison of interviews. One limitation of our semi-structured interviews was possible interviewer bias. Conscious or unconscious, an interviewer input may have influenced respondent answers. Participation bias may have also impacted our results, as participants with an affinity for or interest in antibiotic allergies may be more likely to participate in a study of this type. However, participating interviewees were diverse in terms of gender and experience and accurately represented healthcare providers.

## Conclusion

Incorrect antibiotic allergy registration is a multifactorial and cross-domain problem. The causes are poor registration of symptoms and their duration, insufficient knowledge, lack of awareness and suboptimal communication between healthcare domains and ICT systems. Improving allergy registrations should be an antimicrobial stewardship priority, and interventions should have a domain-transcending approach.

## Supporting information

**S1 Table. Modified checklist of Salden.**
(DOCX)

**S2 Table. Consolidated criteria for reporting qualitative research checklist.**
(DOCX)

**S3 Table. Semi-structured interview.**
(DOCX)

## Acknowledgments

We would like to thank Youssra Atmani (YA) MSc for conducting interviews with Elderly Care healthcare providers and Julia Wubbolts MSc for transcribing interviews verbatim.

## Author Contributions

**Conceptualization:** Martijn Sijbom, Leti van Bodegom-Vos, Bart J. C. Hendriks, Mark G. J. de Boer, Merel M. C. Lambregts.

**Data curation:** Martijn Sijbom, Karolina K. Braun, Bart J. C. Hendriks, Merel M. C. Lambregts.

**Formal analysis:** Martijn Sijbom, Karolina K. Braun, Frederike L. Büchner, Mark G. J. de Boer, Mattijs E. Numans, Merel M. C. Lambregts.

**Investigation:** Martijn Sijbom, Bart J. C. Hendriks, Merel M. C. Lambregts.

**Methodology:** Martijn Sijbom, Leti van Bodegom-Vos, Mark G. J. de Boer, Merel M. C. Lambregts.

**Project administration:** Martijn Sijbom.

**Supervision:** Mark G. J. de Boer, Mattijs E. Numans, Merel M. C. Lambregts.

**Writing – original draft:** Martijn Sijbom, Mark G. J. de Boer, Merel M. C. Lambregts.

**Writing – review & editing:** Martijn Sijbom, Frederike L. Büchner, Leti van Bodegom-Vos, Bart J. C. Hendriks, Mark G. J. de Boer, Mattijs E. Numans, Merel M. C. Lambregts.

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
