## [Decision Letter · Decision Letter 0]

27 Jan 2022

PONE-D-21-34243Cues to improve antibiotic-allergy registration: a mixed-method studyPLOS ONE

Dear Dr. Sijbom,

Thank you for submitting your manuscript to PLOS ONE. After careful consideration, we feel that it has merit but does not fully meet PLOS ONE’s publication criteria as it currently stands. Therefore, we invite you to submit a revised version of the manuscript that addresses the points raised during the review process.

We look forward to receiving your revised manuscript.

Kind regards,

Monika Pogorzelska-Maziarz

Academic Editor

PLOS ONE

Journal Requirements:

2. Thank you for including your ethics statement:  "Before the start of this study, the Ethics Committee of the Leiden University Medical Centre issued a waiver of consent (file number G19.007) to conduct this study.".  

Please amend your current ethics statement to confirm that your named institutional review board or ethics committee specifically approved this study. 

Reviewers' comments:

Reviewer's Responses to Questions

**Comments to the Author**

1. Is the manuscript technically sound, and do the data support the conclusions?

Reviewer #1: Yes

Reviewer #2: Yes

2. Has the statistical analysis been performed appropriately and rigorously? 

Reviewer #1: I Don't Know

Reviewer #2: Yes

3. Have the authors made all data underlying the findings in their manuscript fully available?

Reviewer #1: Yes

Reviewer #2: Yes

4. Is the manuscript presented in an intelligible fashion and written in standard English?

Reviewer #1: Yes

Reviewer #2: Yes

5. Review Comments to the Author

Reviewer #1: Abstract:

Line 42 – Some institutions have reports almost 20% for penicillin specifically, consider including

Introduction

Line 76-77 – consider adding on to the end of this sentence that this is a rare phenomenon and that “over-labeling” occurs

Line 84- Please change “small” to “narrow”

Line 93 – Remove apostrophe from EMRs

Line 93-94 – Do they EMRs link together?

Methods

Line 129-140 – Many times, patients do not know such detailed information about their reactions. How did the authors account for patients who only know they experienced a rash in childhood without other detail? That would be “complete” based on the knowledge available. Can the authors please clarify here?

Line 142-143 – Are cephalosporins prescribed less frequently that all of these other categories?

Line 161-163 – Were there any specialists interviewed or were these all internal medicine / family medicine / hospital medicine focused? Were there any pediatricians? Consider providing more detail here – it comes up in results but would also include here

Results

Table 1 – Remove apostrophe from penicillin’s

Line 198-199 – Without interviewing the patient – it’s difficult to know the records are complete because all of the sought information regarding timing and duration might not be available

Please clarify if all of the information evaluated for in the results is able to be entered in the EMR? If not, these findings aren’t surprising

Table 4 – These quotes are helpful and illuminating

Line 249-251 – Do the primary care physicians and pharmacy records communicate? Or is this “manual” communication, eg faxes or letters?

Discussion

Line 300-302 – But this information is not always known. Wuold make the point that if this information is not available, there should be a way to indicate that so users know the information is complete based on what is available. Please add something to this effect.

Line 302-203 – Would include TEN and DRESS as well

Line 304 – Please add “with the exception of severe cutaneous adverse reactions”

Line 304-306 – Not sure what you’re indicating here – that these types of reactions should not be recorded? Or that effort should be made to distinguish these types of reactions from immediate reactions?

Line 307-309 – This seems redundant – has already been stated. Suggest deleting.

Line 313-314 – please include which members of the healthcare team should be educated? Please include a few more sentences about what they should be educated about regarding hypersensitivity reactions and how to enter them in the EMR.

Line 316 – Change interpreter to “interpret”

Line 336-344 – What else would the authors include in the EMR – similar criteria to the one they used for this study? Most EMRs don’t have this level of detail

Line 345-346 – This is only true for penicillin and less so for other antibiotics. Would delete. Most antibiotic allergy labels are disproved through drug challenges

Line348-350 – This is true for penicillin only. Would add a sentence or two about safety of drug challenges when done by an allergist. There are references that exist in this area

Line 354- Would remove the word “both”

Limitations

Another limitation is that you were unable to verify entered allergy information with patients in the 300 records studied. Also – the level of detail that you prespecified as complete is often not available or not known

Reviewer #2: Very timely, interesting, and well-written manuscript. Research methods clearly described and appropriate use of supporting tables/figures. Just a few brief comments/clarifications/questions:

1. 2% of patients in primary care are allergic - appears reference refers to beta-lactam allergy - this should indicated. I was surprised that only 2% in the outpatient arena - which may be a reflection of my area of practice.

2. line 84 - does small spectrum mean narrow spectrum?

3. again, this may reflect my practice area - why were only elderly care nurses interviews and not those in PCPs? Maybe I misunderstood the sample population?

4. line 316 - interpreter - believe interpret is the correct term?

5. line 329 - it appears allergy information does not communicate between inpatient/acute and outpatient - per line 352. Is that correct? If so, could line 329 be rephrased?

6. line 341 - do you mean PCP should be allowed to remove incorrect/unnecessary allergy information? You are not referring to allergy documentation being removed from the medical record, correct?

I look forward to reading this upon publication - we are attempting a similar project within my facility and I believe this work could be very beneficial to stewardship experts. Thank you for your scientific contributions.

6. PLOS authors have the option to publish the peer review history of their article (what does this mean?). If published, this will include your full peer review and any attached files.

Reviewer #1: No

Reviewer #2: No

---

## [Author Response · Author response to Decision Letter 0]

17 Mar 2022

Dear editor,

We thank the editor and reviewers for the constructive remarks and the editorial board for giving us the opportunity to improve the manuscript. Please find below our responses to the remarks.

Editor

The manuscript has been revised to meet PLOS One’s style requirements.

Please amend your current ethics statement to confirm that your named institutional review board or ethics committee specifically approved this study. 

This issue has been addressed by changing the sentence to: The study was approved by the institutional Ethics Review Board of the Leiden University Medical Center (file number G19.007).

In the rebuttal letter and the revised cover letter, we explain why we cannot share our data freely in a pubic repository, but are available upon reasonable request.

Our study had a mixed method design and was based on two datasets. A set with coded routine Medical Record data, pseudonymized and extracted from primary care practices towards the ELAN datawarehouse, was used for the qualitative analysis of antibiotic allergy registrations. This dataset cannot be shared in an open public repository. Medical data in the ELAN datawarehouse are pseudonymized, so theoretically patients still can be identified and 

confidentiality could be violated. Patients did consent to reuse their medical data for the purpose of dedicated and contextually restricted research and quality management, but not in an open and publicly available domain. So, data are available upon reasonable request at the ELAN datawarehouse through elanresearch.nl. 

The dataset on determinants for incorrect antibiotic allergies registrations cannot be shared in a public repository for other reasons. These data are based on verbatims produced from audio recordings of interviews with research participants. We do not have consent to disseminate the full transcripts and we have to respect the anonymity of the research participants. Although anonymized, participants could be identified through their answers and confidentiality could be violated. Data is available upon reasonable request by emailing the corresponding author, Martijn Sijbom; m.sijbom@lumc.nl.

Reviewer #1

Abstract:

Line 42 – Some institutions have reports almost 20% for penicillin specifically, consider including

This has been changed in the abstract and added in the introduction.

Introduction

Line 76-77 – consider adding on to the end of this sentence that this is a rare phenomenon and that “over-labeling” occurs

We agree with the reviewer that the fact that life threatening reactions are a rare phenomenon should be mentioned. The word ‘rare’ has been added. The word ‘over-labelling’ has not been added as this phenomenon is discussed elsewhere (in the discussion section). 

Line 84- Please change “small” to “narrow”

This has been changed.

Line 93 – Remove apostrophe from EMRs

This has been changed.

Line 93-94 – Do they EMRs link together?

EMRs of primary care and pharmacies are linked together and they exchange medical information such as antibiotic allergy registration. However, the primary care antibiotic allergy label is not electronically linked to the EMR in hospitals nor long term care facilities. Information on allergy labels between primary care and hospitals/long term care facilities is exchanged through referral letters. We describe this in the methods section.

Methods

Line 129-140 – Many times, patients do not know such detailed information about their reactions. How did the authors account for patients who only know they experienced a rash in childhood without other detail? That would be “complete” based on the knowledge available. Can the authors please clarify here?

Our quality analysis was completely conducted with the information available in the antibiotic allergy registration of an EMR. We agree with the reviewer that details of a reaction are often not remembered by the patient, especially when the reaction occurred in remote childhood. Indeed, an incomplete allergy registration may be as ‘complete as possible’ because information is missing because the patient does not remember or is unable to communicate. For decision making however, the registration is incomplete. 

Our aim was to assess the quality of the allergy registrations and secondly identify the potential causes. These causes include patient factors, including the patient not remembering (see also Table 4, patient knowledge). With the determinant “patient knowledge” we are referring to knowledge on antibiotics in general and knowledge of the details of their specific reaction. Based on the reviewers comment we think it is more clear to explicitly describe both aspects of knowledge and have adjusted this in both the table and the text. 

Line 142-143 – Are cephalosporins prescribed less frequently that all of these other categories?

Yes, cephalosporins are hardly prescribed in Dutch primary care. Dutch primary care guidelines on infectious diseases do not recommend the prescription of cephalosporins 

Line 161-163 – Were there any specialists interviewed or were these all internal medicine / family medicine / hospital medicine focused? Were there any pediatricians? Consider providing more detail here – it comes up in results but would also include here

We agree and provide more details on the speciality of the medical specialist in the results section. 

Results

Table 1 – Remove apostrophe from penicillin’s

This has been changed.

Line 198-199 – Without interviewing the patient – it’s difficult to know the records are complete because all of the sought information regarding timing and duration might not be available

Please clarify if all of the information evaluated for in the results is able to be entered in the EMR? If not, these findings aren’t surprising

All relevant information can be entered in the EMR of primary care physicians, we added a sentence to the article clarify this in the method section.

Table 4 – These quotes are helpful and illuminating

Thank you

Line 249-251 – Do the primary care physicians and pharmacy records communicate? Or is this “manual” communication, eg faxes or letters?

EMRs from PCP and pharmacy communicate mainly with each other automatically, they are linked. This has been added. Communication with hospitals and long term care facilities is through referral letters. We describe this in the methods section.

Discussion

Line 300-302 – But this information is not always known. Would make the point that if this information is not available, there should be a way to indicate that so users know the information is complete based on what is available. Please add something to this effect.

We agree and have added: When the details of the reaction can’t be retrieved, for example if the patient does not remember and there is no documentation, this should be indicated in the regsitration. 

Line 302-203 – Would include TEN and DRESS as well

This has been included

Line 304 – Please add “with the exception of severe cutaneous adverse reactions”

This has been added.

Line 304-306 – Not sure what you’re indicating here – that these types of reactions should not be recorded? Or that effort should be made to distinguish these types of reactions from immediate reactions?

The latter, efforts should be made to register all symptoms, their onset and duration and also diagnose the type of reaction. So in the future, other healthcare providers are able to make an informed decision on the prescription of antibiotics. We have added a sentence for clarity. 

Line 307-309 – This seems redundant – has already been stated. Suggest deleting.

We agree, this has already has been stated and therefore deleted.

Line 313-314 – please include which members of the healthcare team should be educated? Please include a few more sentences about what they should be educated about regarding hypersensitivity reactions and how to enter them in the EMR.

We suggest to educate healthcare providers who register antibiotic allergies, which has been added to the manuscript. As this differs per country, we did not suggest specific healthcare providers.

Line 316 – Change interpreter to “interpret”

This has been changed.

Line 336-344 – What else would the authors include in the EMR – similar criteria to the one they used for this study? Most EMRs don’t have this level of detail

Most Dutch EMRs have the possibility to enter in detail the symptoms of allergic reaction. But this is time-consuming and difficult. We have added which information should be registered. We suggest to register all symptoms including their onset and duration, so in the future other healthcare workers can make informed decisions about prescriptions of antibiotic allergies.

Line 345-346 – This is only true for penicillin and less so for other antibiotics. Would delete. Most antibiotic allergy labels are disproved through drug challenges Line348-350 – This is true for penicillin only. Would add a sentence or two about safety of drug challenges when done by an allergist. There are references that exist in this area

We totally agree with the author. Most registrations can be removed based on history alone with or without challenge. Furthermore validated skin tests are not available for all antibiotics. We have removed the paragraph on testing.

We have added: 

Most incorrect antibiotic allergy registrations can be safely removed with a thorough history with or without a provocation test. In most cases, skin testing is not needed. Guidelines on the clinical approach of a potential antibiotic allergy and removing of incorrect antibiotic allergy regsitrations are highly needed. 

Line 354- Would remove the word “both”

This has been removed.

Limitations

Another limitation is that you were unable to verify entered allergy information with patients in the 300 records studied. Also – the level of detail that you prespecified as complete is often not available or not known

It is true we were not available to verify the allergy information in the EMRs, but this was also not the objective of our study. We aimed to assess the quality of a registration in daily medical practice and how this can be improved.

Reviewer #2

Very timely, interesting, and well-written manuscript. Research methods clearly described and appropriate use of supporting tables/figures. Just a few brief comments/clarifications/questions:

Thank you very much for your appreciative comments.

1. 2% of patients in primary care are allergic - appears reference refers to beta-lactam allergy - this should indicated. I was surprised that only 2% in the outpatient arena - which may be a reflection of my area of practice.

This has been changed. In Dutch primary care, the prevalence of antibiotic allergies is lower due to a relatively low prescription rate of antibiotics. In countries where more antibiotics are prescribed, more antibiotic allergies are found, for example, research from the United Stated reported a 15 to 20% prevalence of antibiotic allergies in primary care.

2. line 84 - does small spectrum mean narrow spectrum?

Yes, this has been changed.

3. again, this may reflect my practice area - why were only elderly care nurses interviews and not those in PCPs? Maybe I misunderstood the sample population?

In primary care, specialized nurses do not treat infectious diseases and therefore not prescribe antibiotics. They are specialized in treating chronic conditions such as diabetes mellitus and hypertension. In elderly care, specialized nurses treat infectious diseases and may prescribe antibiotics. 

4. line 316 - interpreter - believe interpret is the correct term?

Yes this has been changed.

5. line 329 - it appears allergy information does not communicate between inpatient/acute and outpatient - per line 352. Is that correct? If so, could line 329 be rephrased?

EMRs from PCP and pharmacy communicate mainly with each other automatically, they are linked. This has been added. Communication with hospitals and long term care facilities is through referral letters. This has been changed and added throughout the manuscript.

6. line 341 - do you mean PCP should be allowed to remove incorrect/unnecessary allergy information? You are not referring to allergy documentation being removed from the medical record, correct?

Sentences on removal of antibiotic allergies from an EMR has been deleted from the manuscript. As this situation is too specific for the Dutch medical infrastructure of EMRs and does not apply to other counties.

---

## [Editor Report · Decision Letter 1]

22 Mar 2022

Cues to improve antibiotic-allergy registration: a mixed-method study

PONE-D-21-34243R1

Dear Dr. Sijbom,

We’re pleased to inform you that your manuscript has been judged scientifically suitable for publication and will be formally accepted for publication once it meets all outstanding technical requirements.

Kind regards,

Monika Pogorzelska-Maziarz

Academic Editor

PLOS ONE

---

## [Editor Report · Acceptance letter]

29 Mar 2022

PONE-D-21-34243R1 

Cues to improve antibiotic-allergy registration: a mixed-method study 

Dear Dr. Sijbom:

I'm pleased to inform you that your manuscript has been deemed suitable for publication in PLOS ONE. Congratulations! Your manuscript is now with our production department. 

Kind regards, 

on behalf of

Dr. Monika Pogorzelska-Maziarz 

Academic Editor

PLOS ONE